# Newborn Screening for SCID. Experience in Spain (Catalonia)

**DOI:** 10.3390/ijns7030046

**Published:** 2021-07-20

**Authors:** Ana Argudo-Ramírez, Andrea Martín-Nalda, Jose Manuel González de Aledo-Castillo, Rosa López-Galera, Jose Luis Marín-Soria, Sonia Pajares-García, Mónica Martínez-Gallo, Marina García-Prat, Roger Colobran, Jacques G. Riviere, Yania Quintero, Tatiana Collado, Antonia Ribes, Judit García-Villoria, Pere Soler-Palacín

**Affiliations:** 1Inborn Errors of Metabolism Division, Biochemistry and Molecular Genetics Department, Hospital Clínic, 08028 Barcelona, Spain; gonzalezde@clinic.cat (J.M.G.d.A.-C.); rmlopez@clinic.cat (R.L.-G.); jlmarin@clinic.cat (J.L.M.-S.); spajares@clinic.cat (S.P.-G.); yquintero@clinic.cat (Y.Q.); collado@clinic.cat (T.C.); aribes@clinic.cat (A.R.); jugarcia@clinic.cat (J.G.-V.); 2Pediatric Infectious Diseases and Immunodeficiencies Unit, Hospital Universitary Vall d’Hebron, Jeffrey Modell Diagnostic and Research Center for Primary Immunodeficiencies, Universitat Autònoma de Barcelona, 08028 Barcelona, Spain; andmarti@vhebron.net (A.M.-N.); marinagarcia@upiip.com (M.G.-P.); jriviere@vhebron.net (J.G.R.); psoler@vhebron.net (P.S.-P.); 3Biomedical Research Institute, August Pi i Sunyer (IDIBAPS), 08036 Barcelona, Spain; 4Spain Center for Biomedical Research Network on Rare Diseases (CIBERER), 28029 Madrid, Spain; 5Immunology Division, Hospital Universitary Vall d’Hebron, Jeffrey Modell Diagnostic and Research Center for Primary Immunodeficiencies, Universitat Autònoma de Barcelona, 08028 Barcelona, Spain; mmartinez@vhebron.net (M.M.-G.); rcolobran@vhebron.net (R.C.); 6Department of Clinical and Molecular Genetics, Hospital Universitari Vall d’Hebron, Jeffrey Modell Diagnostic and Research Center for Primary Immunodeficiencies, Universitat Autònoma de Barcelona, 08028 Barcelona, Spain

**Keywords:** newborn screening, severe combined immunodeficiency, T-cell receptor excision circles, T-lymphocytes, stem cell transplantation, PNP deficiency, TREC, SCID, NBS Spain, NBS Catalonia

## Abstract

Newborn screening (NBS) for severe combined immunodeficiency (SCID) started in Catalonia in January-2017, being the first Spanish and European region to universally include this testing. In Spain, a pilot study with 5000 samples was carried out in Seville in 2014; also, a research project with about 35,000 newborns will be carried out in 2021–2022 in the NBS laboratory of Eastern Andalusia. At present, the inclusion of SCID is being evaluated in Spain. The results obtained in the first three and a half years of experience in Catalonia are presented here. All babies born between January-2017 and June-2020 were screened through TREC-quantification in DBS with the Enlite Neonatal TREC-kit from PerkinElmer. A total of 222,857 newborns were screened, of which 48 tested positive. During the study period, three patients were diagnosed with SCID: an incidence of 1 in 74,187 newborns; 17 patients had clinically significant T-cell lymphopenia (non-SCID) with an incidence of 1 in 13,109 newborns who also benefited from the NBS program. The results obtained provide further evidence of the benefits of early diagnosis and curative treatment to justify the inclusion of this disease in NBS programs. A national NBS program is needed, also to define the exact SCID incidence in Spain.

## 1. Introduction

Severe combined immunodeficiency (SCID), the most severe form of T-cell primary immunodeficiency (PID), includes a group of inherited defects characterized by severe T-cell lymphopenia (TCL). Patients with SCID require prompt clinical intervention to prevent severe life-threatening infections, and several studies have reported significantly improved survival in babies diagnosed at birth [1,2]. Curative treatment is based on hematopoietic stem cell transplantation (HSCT) or gene therapy when available [2]. SCID can be screened at birth in a cost-effective way on a large scale through quantification of T-cell receptor excision circles (TREC) in Guthrie card dried blood spot (DBS) samples [3,4].

Since its initial implementation in Wisconsin in 2008, NBS for SCID using a T-cell receptor excision circle (TREC) assay has been established worldwide. SCID was officially included in the Catalonian NBS program in January 2017 [5], being the first NBS program in Spain and Europe.

### 1.1. Newborn Screening (NBS) Programs in Spain

The territorial distribution of Spain is made up of seventeen autonomous regions plus two autonomous cities located in Africa, with large inter-region variability in terms of land area and number of births. NBS is organized into fifteen laboratories around the country [6].

The Spanish Government sets a mandatory main screening panel that includes seven diseases (Table 1). However, each region has its own competence and devolved powers in Health, so each region can decide to include more diseases in their NBS programs. Consequently, the majority of regions have more than seven diseases included. Some regions include only the minimum of 7 diseases, where others include up to 28, as for example is the case in Galicia, 26 in Murcia, or 24 in Catalonia. Furthermore, some regions, as Catalonia, report the incidental findings (secondary panel) of other diseases [7], although these are not included in the official panel [6].

### 1.2. Newborn Screening for SCID in Spain

At present, the only region that has officially included SCID in its program is Catalonia, since January 2017 [5]. A pilot study with 5000 samples was carried out in Seville in 2014 [8]. Moreover, a research project with about 35,000 newborns will be carried out in 2021–2022 for both SCID and Spinal muscular atrophy (SMA) in the NBS laboratory of Eastern Andalusia [9].

### 1.3. Newborn Screening in Catalonia

The NBS program in Catalonia began more than 50 years ago with phenylketonuria (1969) and has increased to up to 24 diseases in 2017 with the inclusion of SCID [10]. Since 2015, the NBS laboratory has been accredited according to the quality standards ISO 15189, and since 2018, SCID detection is included in this accreditation. Nowadays, around 60,000 newborns per year are screened for the 24 diseases included in the Catalonian NBS program, which encompasses phenylketonuria, congenital hypothyroidism, cystic fibrosis, sickle cell disease, amino acid, organic acid and mitochondrial beta-oxidation disorders, and, lately, SCID (Table 1). Incidental findings found throughout differential diagnosis of the main panel are also reported.

### 1.4. Newborn Screening for SCID in Catalonia

In September 2016, the Department of Public Health of the Catalonian Government officially communicated the approval of SCID inclusion in its NBS program, being the first program in Europe to do so. In January 2017, NBS for SCID in Catalonia started a six months prospective implementation pilot study [5]. Regular verbal informed consent for NBS from parents was obtained from all participants included in the study; a specific written informed consent whose genetic evaluation was needed was also obtained. Data from the experience of the first three and a half years following its implementation are presented here.

## 2. Materials and Methods

### 2.1. Population

From 1 January to 30 June 2017, all consecutive dried blood spot (DBS) samples received as part of the universal NBS program in Catalonia (*n* = 33,040) underwent SCID screening as part of a six-month prospective implementation pilot study to validate our approach. DBS samples received from all newborns born between January 2017 and June 2020 were analyzed (*n* = 222,857).

### 2.2. Sample Testing

Quantification of TRECs in DBS (1.5 mm diameter spot) was performed according to the commercial Enlite Neonatal TREC kit instructions (PerkinElmer, Turku, Finland). The EnLite TREC kit is a combination of PCR-based nucleic acid amplification and time-resolved fluorescence resonance energy transfer (TR-FRET) based detection. The assay detects two targets simultaneously: TRECs and beta-actin, which is used as the internal control to monitor specimen amplification in each test. It basically consists of three steps: firstly, DNA elution; secondly, TREC and beta-actin gene, both amplification and hybridization and lastly, signal measurement with a Victor^®^ photometer.

### 2.3. Pilot Study

#### Methodology and Algorithm Validation

We decided to validate our methodology with the solution offered by Perkin Elmer, which included all the equipment, reagents and specific software necessary to analyze the samples and perform the interpretation of results. Quantification of TREC in DBS (1.5 mm diameter spot) was performed according to the commercial Enlite Neonatal TREC kit instructions (PerkinElmer, Turku, Finland). The kit is a combination of PCR-based nucleic acid amplification and time-resolved fluorescence resonance energy transfer (TR-FRET) based detection. The assay detects two targets simultaneously: TRECs and beta-actin, which is used as the internal control to monitor specimen amplification in each test.

The method was verified (repeatability, reproducibility, limit of detection, limit of quantification, sensitivity and specificity), and these parameters were successfully compared with those stated in the kit insert [5]. In addition, available NBS samples from children with a known SCID diagnosis in Catalonia in the previous five years were analyzed as positive controls (*n* = 6; median, range TREC copies/µL: 2, 2–4), as well as five other positive samples from the SCID Newborn Screening Quality Assurance Program-Proficiency Testing Program provided by the CDC (Centers for Disease Control and Prevention, Atlanta, GA, USA).

After reviewing the decision algorithms from other NBS programs with previous experience on this disease, we decided to start the pilot study with the algorithm used by Audrain et al. [11], which had a retest cutoff of 34 copies/μL and detection cutoff of 20 copies/µL [5].

The retest after the first sample rate (retest rate), requested second sample rate and SCID-positive detection rate (positive detections) were calculated. Based on these results, the algorithm was reevaluated after the first year of experience (*n* = 66,811).

### 2.4. Currently Decision Algorithm for SCID Detection

The current decision algorithm is shown in Figure 1. From 2018, we have been using 24 copies/µL of TREC as retest cutoff and 20 copies/µL as detection cutoff (always with beta-actin above 50 copies/µL). Besides, we have set alarm cutoffs depending on the newborn’s gestational age: term newborns with a value of 10 copies/µL or less and preterm newborns with 5 copies/µL or less are referred directly to the hospital as a positive screening. A second sample is requested if TREC values are between these alarm levels and 20 copies/µL. If TREC values in the second sample remain lower than 20 copies/µL, the newborn is also referred to the hospital.

### 2.5. Protocol for SCID Positive Detections

All positive cases referred to the hospital are visited at the Reference Clinical Unit as soon as possible (always within the first seven days after detection), where clinical and immunological assessment is performed per protocol (Figure 2). Complete family and medical charts are recorded, and a meticulous physical examination is carried out. In addition, psychological support is offered to parents, starting at their first visit.

Several tests are performed (Figure 2): T, B and NK cell immunophenotyping, expression of different markers on T lymphocytes, in vitro lymphocyte proliferation and immunoglobulins counts in order to confirm or discard SCID criteria.

In cases with non-SCID lymphopenia, a consultation is scheduled with the geneticist for clinical evaluation and CGH-array studies. If these results are normal, a genetic custom-designed next-generation sequencing (NGS) based panel that targets 323 genes, including most of the known PID-causing genes, is performed.

## 3. Results

In July 2017, NBS for SCID officially started in Catalonia, and in 2019 we published our two years of experience [5]. Now, after three and a half years, we have analyzed 222,857 samples from Catalonian newborns, obtaining a retest rate of 2%, a second sample requested rate of 0.2% (*n* = 470) and a positive detection rate of 0.02%, with a total of 48 positive detections referred to the hospital (Figure 3).

Comparative results of SCID NBS are shown in Table 2. After reducing our retest cutoff from 34 to 24 copies/µL, the retest rate decreased from 3.3% in 2017 to 1.4% in 2018, with similar results in the following years (and similar retest rates published by others [12]). That led to an important reduction (40%) in the number of samples needing to be repeated (in duplicate). Moreover, the evolution of the incidence regarding the increasing number of newborns analyzed is shown.

The final diagnosis of the 48 positive detections is shown in Figure 4. There have been three cases of SCID (0, 0 and 4 TREC copies/µL, respectively) with confirmed diagnosis: in case 1, we could not find causal mutations; in case 2 a RAG2 deficiency was found; a PNP deficiency was found in case 3; establishing an incidence of SCID in Catalonia of one in 74,187 newborns. All three patients underwent hematopoietic stem cell transplantation (HSCT) at the age of two months of life using reduced-intensity conditioning, with a good clinical outcome and immunological reconstitution.

The remaining cases were classified as follows: seventeen cases with non-SCID lymphopenia (median, range TREC copies/µL; 10 (5–18)) with an incidence of one case in 13,109 newborns: eight cases of 22q11del syndrome, four idiopathic lymphopenia, two congenital chylothorax, two cases of prematurity and one Down syndrome. One newborn had transient lymphopenia due to an initially low lymphocyte count with recovery in the following months; twenty-two patients were considered false-positive cases (median, range TREC copies/µL; 13 (8–20)) because of an initially normal lymphocyte count with normalization of TREC between three and six months of life; and five patients are still under study (one of them is a suspected leaky-SCID case).

## 4. Discussion

This paper reports the current situation of NBS for SCID in Spain and the data obtained from the Catalonian experience (January 2017–June 2020). The SCID detection based on TREC quantification enables detection of non-SCID lymphopenic conditions in addition to SCID. After three and a half years of experience and almost 223,000 newborns screened, 3 SCID patients and 17 other conditions have been identified. Two SCID cases were from consanguineous Moroccan parents, of which one of them had a history of a sister who died in Morocco with described symptoms compatible with Omenn-type SCID. Three patients underwent HSCT at the age of two months of life using reduced-intensity conditioning, with a good clinical outcome and immunological reconstitution. To the best of our knowledge, no cases of SCID or clinically significant T-cell lymphopenia have been missed in our screened cohort. In addition, in the Catalonian NBS program, since November 2019, the quantification of the concentration of adenosine and deoxyadenosine in DBS (NeoBase 2 Non-derivatized MSMS kit, Perkin Elmer) is performed; adenosine deaminase (ADA) deficiency cases that could not have been detected by the TREC method would be detected with this methodology. The detection strategy and the protocol for characterization, diagnosis and follow-up of positive cases are now consolidated.

We implemented the commercially available EnLite Neonatal TREC kit (Perkin Elmer, Turku, Finland), which has been validated elsewhere [11,13,14,15]. This kit allows for DNA elution and gene amplification in a single process, as well as standardization and reproducibility according to ISO 15189 standards. The initial retest cutoff was 34 copies/µL (3rd percentile), which was changed to 24 copies/µL (1st percentile) in 2018 because our retest rate was too high when compared to published data [12], whereas the request for second sample cutoff was maintained at 20 copies/µL (always with beta-actin above 50 copies/µL). We plan to reevaluate our data again after five years of experience in order to further reduce the established cutoffs, and therefore the retest rates, to improve even more the overall efficacy of the program.

In conclusion, TREC quantification in DBS for SCID detection has been satisfactorily implemented in the Catalonian NBS program, fulfilling the ISO 15189 quality standards. All retested, requested second samples, and positive detection rates are optimal with the current algorithm and similar to published data [12]. SCID detection strategy and diagnosis and follow-up protocol have been consolidated. After three and a half years of experience with 222,857 newborns screened, three patients with SCID have been identified and another seventeen patients with other causes of lymphopenia have also benefited from the NBS program. The incidence of SCID in Catalonia is one in 74,187 newborns, which could be higher if the leaky-SCID case is confirmed. The incidence of non-SCID lymphopenia is 1 in 13,109. We will reevaluate our data after five years of experience in order to further reduce our retesting rates.

The results obtained provide further evidence of the benefits of early diagnosis and curative treatment to justify the inclusion of this disease in NBS programs in other regions and countries. It seems reasonable to start NBS for SCID in Spain and to reevaluate its usefulness after two to three years, with a sufficient number of patients to know its true incidence in the country. A joint collaboration between governmental agencies, NBS laboratories and clinical units is key to achieve it.

## Figures and Tables

**Figure 1 IJNS-07-00046-f001:**
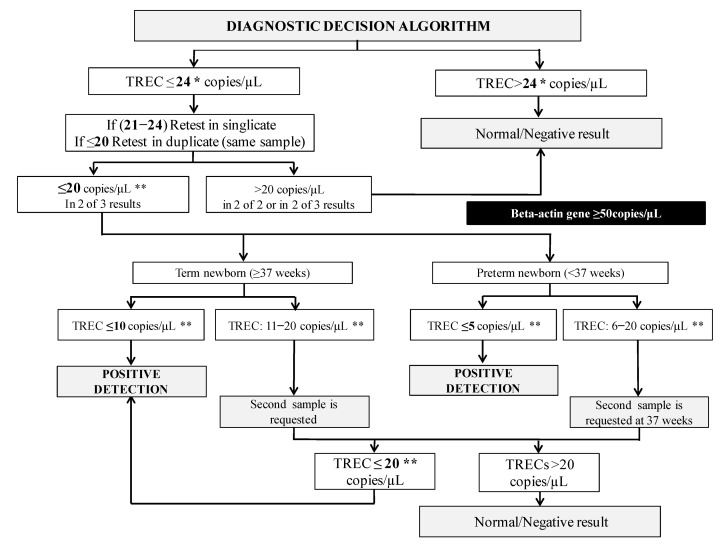
SCID NBS detection decision algorithm. * The retest cutoff was changed from 34 to 24 copies/µL in 2018. ** If the beta-actin gene < 50 copies/µL, a second sample is requested because the sample was considered of unsatisfactory quality. Abbreviations: TREC: T-cell receptor excision circles.

**Figure 2 IJNS-07-00046-f002:**
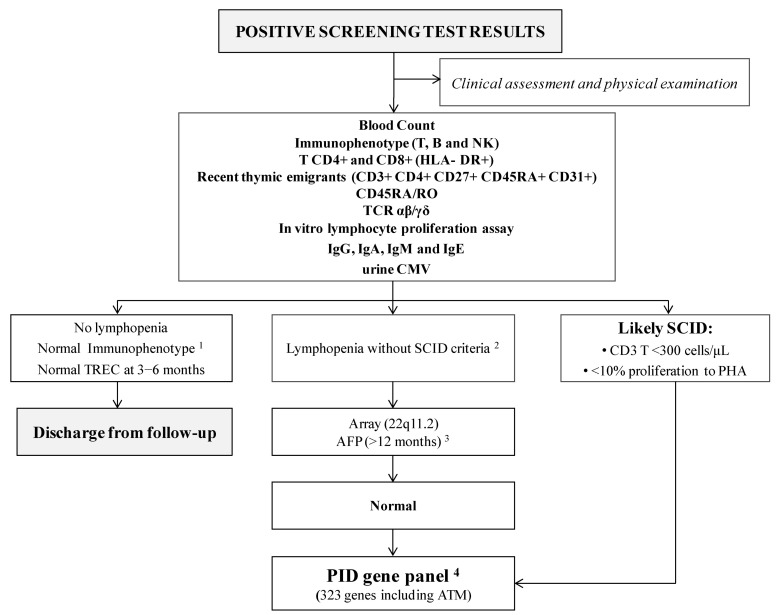
Immunological and genetic protocol in positive cases. ^1^ Flow cytometry protocols are shown in [5]; ^2^ Lymphopenia and SCID criteria are shown in [5]; ^3^ AFP values are not reliable before 12 months of age; therefore, AFP was only studied in patients older than one year lacking an alternative diagnosis; ^4^ The list of 323 PID genes is shown in [5]. Abbreviations: AFP, alpha-fetoprotein; ATM, ataxia telangiectasia; B, B-cells; CMV, cytomegalovirus; Ig, immunoglobulin; NK, natural killer cells; PID, Primary immunodeficiency; SCID, severe combined immunodeficiency; T, T-cells; TCR, T-cell receptor; TRECs, T-cell receptor excision circles.

**Figure 3 IJNS-07-00046-f003:**
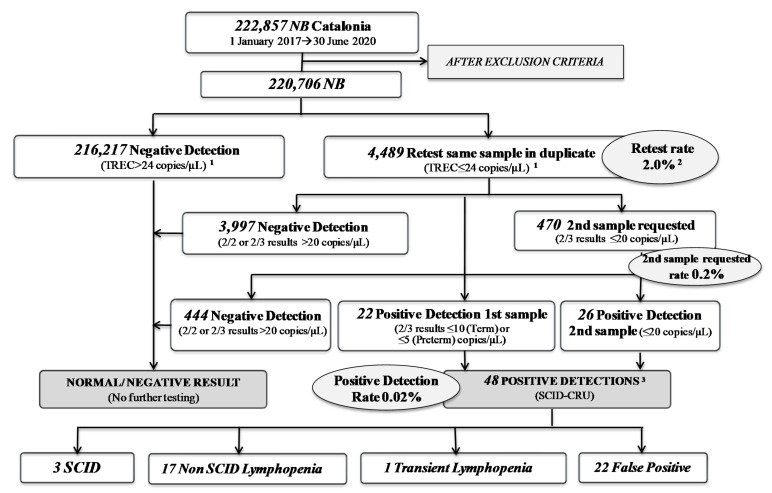
Results of SCID NBS in Catalonia over the study period (January 2017–June 2020). ^1^ 2018 cutoff (2017 cutoff = 34 copies/µL); ^2^ Total period—Retest rate (2017—Retest rate = 3.34% (cutoff = 34 copies/µL); 2018-2020 Retest rate = 1.4% (cutoff = 24 copies/µL); ^3^ Five patients are currently under study. (Note: Exclusion criteria considered: collection time before 44 h or after seven days of life, transfusions, poor DNA amplification, and poor quality or blood amount). Abbreviations: NB, newborns; SCID, severe combined immunodeficiency; SCID-CRU, SCID Clinical Reference Unit.

**Figure 4 IJNS-07-00046-f004:**
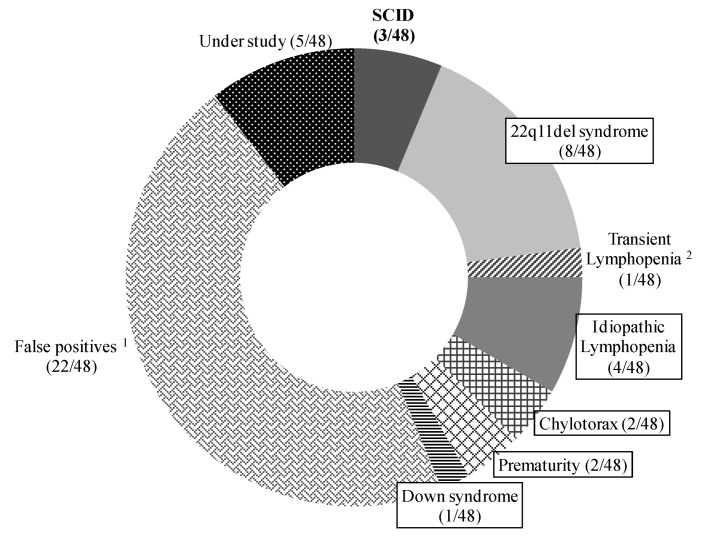
Final diagnoses in patients testing positive in Catalonia (January 2017–June 2020). ^1^ False-positives due to initially normal lymphocyte count with normalization of TRECs between three and six months of life; ^2^ Transient lymphopenia due to initially low lymphocyte count with recovery in the following months. Note: Final diagnoses of the 17 non-SCID lymphopenic patients are boxed. Abbreviations: SCID, severe combined immunodeficiency.

**Table 1 IJNS-07-00046-t001:** Diseases included in the newborn screening program of Catalonia.

Inherited Metabolic Diseases	
Phenylketonuria and Hyperphenyalaninemias	ß-Ketothiolase deficiency
Maple syrup urine disease	Propionic acidemia
Tyrosinemia type I	**Medium-chain** **acyl-CoA dehydrogenase deficiency**
Citrullinemia	Very long-chain acyl-CoA dehydrogenase deficiency
Homocystinuria (CBS deficiency)	**Mitochondrial trifunctional protein deficiency//Long-chain L-3 hydroxyacyl-CoA dehydrogenase deficiency**
**Glutaric aciduria type I**	Multiple acyl-CoA dehydrogenase deficiency
Isovaleric acidemia	Primary carnitine deficiency
Methylmalonic aciduria	Carnitine palmitoyltransferase 1 deficiency
Methylmalonic aciduria with Homocystinuria	Carnitine palmitoyltransferase 2 deficiency
3-Hydroxy-3-methylglutaryl-CoA lyase deficiency	Carnitine-acylcarnitine translocase deficiency
**Other diseases**	
**Cystic Fibrosis**	**Sickle cell disease**
**Congenital Hypothyroidism**	Severe combined immunodeficiency

Seven diseases included in the main screening panel in Spain are highlighted in bold.

**Table 2 IJNS-07-00046-t002:** Annual comparative data of NBS for SCID in Catalonia.

PARAMETER	2020 (January–June)	2019	2018	2017
Sample size	30,296	61,460	64,290	66,811
Repetition Cutoff (same sample in duplicate)	24 copies/µL	24 copies/µL	24 copies/µL	34 copies/µL
1st Sample Retest Rate	1.5% (*n* = 464)	1.5% (*n* = 915)	1.4% (*n* = 898)	3.34% (*n* = 2212)
2nd Sample Request Rate	0.19% (*n* = 57)	0.17% (*n* = 108)	0.26% (*n* = 167)	0.21% (*n* = 138)
Positive Detection Rate	0.02% (*n* = 7)	0.02% (*n* = 11)	0.02% (*n* = 15)	0.02% (*n* = 15)
Incidence	1/74,187 *	1/193,002	1/130,903	-

* Calculated with the total newborns in Catalonia between 1 January 2017 and 30 June 2020.

## Data Availability

The data are not publicly posted, and can be requested by contacting the corresponding author.

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
