# Peer review of "Newborn Screening for SCID. Experience in Spain (Catalonia)"

_2409-515X, 2021, doi:10.3390/ijns7030046_

Round 1
Reviewer 1 Report
In this paper, the authors present a good summary of results of their newborn screening program for SCID, where 222,857 newborns were screened using TREC quantification over a 3.5 year period. This is a follow-up paper, reporting additional data since the group’s publication of their 2 year experience. This data enables the incidence of SCID and T cell lymphopaenia in Catalonia to be reported, and provides data regarding cut-off values and recall rates in a large number of screened infants.
Overall, this paper is well written, however please refer to highlighted areas in the pdf file where attention to sentence structure/phrasing/grammar/typographical errors is required.
Further specific comments are below.
Introduction
The introduction lacks some detail in terms of the relevant background, and begins with a discussion of NBS programs in Spain and Catalonia. It is suggested that the introduction is expanded to include more background information on the topic, i.e. a few statements on PID/IEI, SCID and its definition, why screening is important and the rationale for including SCID as a targeted screening condition in NBS programs. A few lines on TRECs and how they are measured would also improve this section. The discussion on NBS programs in Spain could then follow these more general background statements.
Methodology
Please expand on the methodologies used for TREC measurement, only a brief summary is provided here.
In the discussion, you refer to the literature regarding the recall rate for re-testing, but cite only one paper published in 2015 (10). Are there any newer/additional publications using this methodology to which you can compare your recall/re-testing rates?
Consent
A comment is made that consent was obtained where genomic testing was performed (this line also requires review of sentence structure). Please comment further on the consent process for NBS in Catalonia (it sounds as though there is an ‘opt-out’ rather than ‘opt-in’ process for NBS). Please clarify in the text.
Tables & Figures
Table 1 – suggest grouping the seven common screened diseases together, or highlight in bold or italics to increase clarity in the figure. It takes some time for the reader to find the superscript numbers.
Figure 1 – Correct typo ‘Beta-actin gene’, capital B, add ‘e’ to gene
Figure 2 – reword top box to ‘Positive screening test result’
Change ‘stop follow-up’ to ‘Discharge from follow-up’
Change ‘SCID suspicion’ to ‘Likely SCID’
Include ‘clinical assessment and physical examination’ in the algorithm (this is mentioned in the text but would be good to add to the figure).
Figure 3 – what were the exclusion criteria mentioned in the top part of the table? Why were 2151 newborns excluded? Please explain.
Additional Table:
- It would be useful to provide further detailed information regarding the 21 cases identified (with SCID, T cell lymphopaenia due to other causes & transient lymphopaenia).
- Please add another table showing further details of these cases identified during the screening program. This could include information including the TREC levels, gestational age, diagnosis (e.g. broad diagnosis ‘SCID’, specific diagnosis ‘PNP deficiency’, molecular diagnosis (provide the specific details of the genetic mutation, if known), and salient laboratory findings (e.g. CD3+ T cell count, naïve T cell count etc), clinical progress/outcomes (e.g. HSCT, conservative management etc).
- The case mentioned of likely leaky SCID is also of particular interest and should be included in this table with as many details as currently available (is there a molecular diagnosis that can be reported? If this can be confirmed prior to publication, as you mention, your SCID incidence will be even higher)
Comment further on the 22 ‘false positive’ results in the text. How long have these children been followed up for? Did any of the children have follow-up since normalisation of their lymphocyte counts at 3-6 months? Have any of the children since developed symptoms suggesting an underlying immunodeficiency? Are there more details regarding the 5 patients still under evaluation/clinical follow-up (include details in table above)? (suggest changing the phrasing ‘under study’ to ‘under clinical evaluation’)
Are the authors aware of any cases of SCID/T cell lymphopaenia that have been missed in the screening program to date (any cases that returned an initially negative (normal) screening test, but have since been diagnosed with one of these conditions? It is acknowledged that TREC-based screening has some limitations and will identify most, but not all cases of SCID. If so, please provide details, if not, perhaps make a statement to this effect – e.g. ‘to the best of our knowledge, no cases of SCID or clinically significant T cell lymphopaenia have been missed in our screened cohort.’
Perhaps expand the discussion further regarding clinical outcomes for identified cases, either in the text and/or table. Your final statement in the abstract and conclusion refers to ‘providing clinical evidence of the benefits of early diagnosis and treatment’, hence this discussion should be expanded in the text.

Author Response
Thank you for the proposal and for all the comments and corrections provided. We have preferred not include some specific suggestions/comments from reviewers in the article, since the objective of this publication was to summarize the communication made during the virtual meeting “Newborn Screening for SCID ‘State of the Art’” celebrated on 26 and 27 January 2021. We know some information is missing but we are also working on a more complete publication on the 5 years of neonatal screening for SCID in Catalonia and we will take into account all the comments made. We are very grateful for the excellent review work done and for the time spent reading our paper. We specify in italics each of the sections in detail below.
REVIEWER 1
Introduction
The introduction lacks some detail in terms of the relevant background, and begins with a discussion of NBS programs in Spain and Catalonia. It is suggested that the introduction is expanded to include more background information on the topic, i.e. a few statements on PID/IEI, SCID and its definition, why screening is important and the rationale for including SCID as a targeted screening condition in NBS programs. A few lines on TRECs and how they are measured would also improve this section. The discussion on NBS programs in Spain could then follow these more general background statements.
This paragraph has been added into 1. Introduction Section:
“Severe combined immunodeficiency (SCID), the most severe form of T-cell primary immunodeficiency (PID), includes a group of inherited defects characterized by severe T-cell lymphopenia (TCL). Patients with SCID require prompt clinical intervention to prevent severe life-threatening infections, and several studies have reported significantly improved survival in babies diagnosed at birth (1,2). Curative treatment is based on hematopoietic stem cell transplantation (HSCT) or gene therapy when available (2). SCID can be screened at birth in a cost-effective way on a large scale through quantification of T-cell receptor excision circles (TRECs) in Guthrie card dried blood spot (DBS) samples (3,4).
Since its initial implementation in Wisconsin in 2008, NBS for SCID using a T-cell receptor excision circle (TREC) assay has been established worldwide. SCID was officially included in the Catalonian NBS program in January 2017 (7) being the first NBS program in Spain and Europe”
Methodology
Please expand on the methodologies used for TREC measurement, only a brief summary is provided here.
This paragraph has been added into 2.3.1 Methodology Section:
“Quantification of TREC in DBS (1.5 mm diameter spot) was performed according to the commercial Enlite Neonatal TREC kit instructions (PerkinElmer, Turku, Finland). The kit is a combination of PCR-based nucleic acid amplification and time-resolved fluorescence resonance energy transfer (TR-FRET) based detection. The assay detects two targets simultaneously: TRECs and beta-actin, which is used as the internal control to monitor specimen amplification in each test”.
In the discussion, you refer to the literature regarding the recall rate for re-testing, but cite only one paper published in 2015 (14). Are there any newer/additional publications using this methodology to which you can compare your recall/re-testing rates?
“The initial retest cutoff was 34 copies/µL (3rd percentile), which was changed to 24 copies/µL (1st percentile) in 2018, because our retest rate was too high when compared to published data (10)”
When we decided to change our retest cutoff, reference 10 was the only reference available. Although it is a really good suggestion from the author, we think we shouldn’t use later references to justify this change.
Consent
A comment is made that consent was obtained where genomic testing was performed (this line also requires review of sentence structure). Please comment further on the consent process for NBS in Catalonia (it sounds as though there is an ‘opt-out’ rather than ‘opt-in’ process for NBS). Please clarify in the text.
This sentence has been added into 1.4 Newborn screening for SCID in Catalonia Section:
“Regular verbal informed consent for NBS from parents was obtained from all participants included in the study; a specific written informed consent whose genetic evaluation was needed was also obtained”.
Tables & Figures
Table 1 – suggest grouping the seven common screened diseases together, or highlight in bold or italics to increase clarity in the figure. It takes some time for the reader to find the superscript numbers.
Table 1 has been modified, with the seven diseases included in the main screening panel highlighted in bold.
Figure 1 – Correct typo ‘Beta-actin gene’, capital B, add ‘e’ to gene
Figure 1 has been modified, adding “Beta-actin gene”.
Figure 2 – reword top box to ‘Positive screening test result’
Change ‘stop follow-up’ to ‘Discharge from follow-up’
Change ‘SCID suspicion’ to ‘Likely SCID’
Include ‘clinical assessment and physical examination’ in the algorithm (this is mentioned in the text but would be good to add to the figure).
Figure 2 has been modified, adding all the reviewer’s suggestions.
Figure 3 – what were the exclusion criteria mentioned in the top part of the table? Why were 2151 newborns excluded? Please explain.
Figure 3 has been modified. We have added at the foot of the figure the exclusion criteria considered (also included in reference 7).
Additional Table:
- It would be useful to provide further detailed information regarding the 21 cases identified (with SCID, T cell lymphopaenia due to other causes & transient lymphopaenia).
Not included.
- Please add another table showing further details of these cases identified during the screening program. This could include information including the TREC levels, gestational age, diagnosis (e.g. broad diagnosis ‘SCID’, specific diagnosis ‘PNP deficiency’, molecular diagnosis (provide the specific details of the genetic mutation, if known), and salient laboratory findings (e.g. CD3+ T cell count, naïve T cell count etc), clinical progress/outcomes (e.g. HSCT, conservative management etc).
Not included.
- The case mentioned of likely leaky SCID is also of particular interest and should be included in this table with as many details as currently available (is there a molecular diagnosis that can be reported? If this can be confirmed prior to publication, as you mention, your SCID incidence will be even higher)
Not included.
Comment further on the 22 ‘false positive’ results in the text. How long have these children been followed up for? Did any of the children have follow-up since normalisation of their lymphocyte counts at 3-6 months? Have any of the children since developed symptoms suggesting an underlying immunodeficiency? Are there more details regarding the 5 patients still under evaluation/clinical follow-up (include details in table above)? (suggest changing the phrasing ‘under study’ to ‘under clinical evaluation’)
Not included.
Are the authors aware of any cases of SCID/T cell lymphopaenia that have been missed in the screening program to date (any cases that returned an initially negative (normal) screening test, but have since been diagnosed with one of these conditions? It is acknowledged that TREC-based screening has some limitations and will identify most, but not all cases of SCID. If so, please provide details, if not, perhaps make a statement to this effect – e.g. ‘to the best of our knowledge, no cases of SCID or clinically significant T cell lymphopaenia have been missed in our screened cohort.’
This paragraph has been added into 4. Discussion Section:
To the best of our knowledge, no cases of SCID or clinically significant T cell lymphopaenia have been missed in our screened cohort. In addition, in the PCN of Catalonia, since November 2019, the quantification of the concentration of adenosine and deoxyadenosine in DBS (NeoBase 2 Non-derivatized MSMS kit, Perkin Elmer), so that adenosine deaminase (ADA) deficiency cases that were not detected by the TREC method would be detected with this methodology.
Perhaps expand the discussion further regarding clinical outcomes for identified cases, either in the text and/or table. Your final statement in the abstract and conclusion refers to ‘providing clinical evidence of the benefits of early diagnosis and treatment’, hence this discussion should be expanded in the text.
Not included.
As we already mentioned, this publication was to summarize the communication made during the virtual meeting “Newborn Screening for SCID ‘State of the Art’” celebrated on 26 and 27 January 2021. We know some information is missing but we are also working on a more complete publication on the 5 years of neonatal screening for SCID in Catalonia and we will take into account all the comments made.
We really appreciate the great work done by the reviewers and hope they understand our reasons for not incorporating some of the proposed changes.

Reviewer 2 Report
Introduction
Section 1.2 – The previous 5000 sample study utilised a Triplex method and the current methods used by the authors is the EnLite system. Why was there this change of methodology choice?
Which method will be used in the upcoming study that includes SMA?
Section 2.3 “The retest after the first sample rate (retest rate), requested second sample rate, and
SCID-positive detection rate (positive detections) were calculated. Based on these results,
the algorithm was reevaluated.” After how many samples was the re-evaluation?
Section 2.5 – “All positive cases referred to the hospital are visited at the Reference Clinical Unit as
soon as possible….” Can the authors be more specific? For example, what timeframes are there for referral, how are bank holidays accounted for, do they think any extra delay would impact patient care?
Figure 2. – Patients with no lymphopaenia and normal phenotype has normal TRECs at 3-6 months. How was this measured – using bloodspots with same algorithm or with fresh blood and sorted cells?
Figure 3. – “After Exclusion Criteria” – what was this criteria?
Results
“Comparative results of SCID NBS are shown in table 2. After reducing our retest cutoff
from 34 to 24 copies/μL…” - what was the rationale for picking 24 copies/ul and not lower?
Table 2. – What was the delay between the 1st result and the repeat sample arriving and being tested?
Page 7 – “There have been three cases of SCID with confirmed diagnosis…” - what were the TREC levels in these 3 cases? What was the median/range TREC levels for the other screen positive groups?
What were the limits of detection and quantification for this assay based on the authors results?
Discussion
Page 7, “After three and a half years’ experience and almost 223,000 newborns
screened, three SCID patients…” - were these from families with known histories of SCID?
Were any SCID babies missed by screening in Catalonia over the 3.5 years?
Has the incidence of SCID in Catalonia increased since the start of screening?
Would lowering the cut-offs remove non-SCIDs from the positive screen group? If so, could the authors provide details on the numbers in the positive screen group if different cut-offs had been used? For example if the first cut-off was 15 or 20 copies what impact would this have? Maybe a table with various cut-off alongside referral numbers would be helpful.
Author Response
Thank you for the proposal and for all the comments and corrections provided. We have preferred not include some specific suggestions/comments from reviewers in the article, since the objective of this publication was to summarize the communication made during the virtual meeting “Newborn Screening for SCID ‘State of the Art’” celebrated on 26 and 27 January 2021. We know some information is missing but we are also working on a more complete publication on the 5 years of neonatal screening for SCID in Catalonia and we will take into account all the comments made. We are very grateful for the excellent review work done and for the time spent reading our paper. We specify in italics each of the sections in detail below.
REVIEWER 2
Introduction
Section 1.2 – The previous 5000 sample study utilised a Triplex method and the current methods used by the authors is the EnLite system. Why was there this change of methodology choice?
The pilot study with 5,000 samples was carried out in Seville (other region of Spain). This study was independent, and with no relation with NBS Program in Catalonia. Our study pilot was carried out with the same method currently used (Enlite Perkin Elmer)
We have added details about method mentioned in Section 1.2.
Which method will be used in the upcoming study that includes SMA?
As the information is from other NBS program, we prefer not to specify the method, because the study has not officially started yet.
Section 2.3 “The retest after the first sample rate (retest rate), requested second sample rate, and SCID-positive detection rate (positive detections) were calculated. Based on these results,the algorithm was reevaluated.” After how many samples was the re-evaluation?
This evaluation was with 66,811 samples, after one year of experience. We have added this information.
Section 2.5 – “All positive cases referred to the hospital are visited at the Reference Clinical Unit as soon as possible….” Can the authors be more specific? For example, what timeframes are there for referral, how are bank holidays accounted for, do they think any extra delay would impact patient care?
Within the first seven days after detection, all positive cases were referred to SCID-CRU, where clinical and immunological assessment was performed per protocol (Figure 2). We have added this information.
Figure 2. – Patients with no lymphopaenia and normal phenotype has normal TRECs at 3-6 months. How was this measured – using bloodspots with same algorithm or with fresh blood and sorted cells?
Using blood spot samples.
Figure 3. – “After Exclusion Criteria” – what was this criteria?
We have added at the foot of the figure the exclusion criteria considered (also included in reference 3).
Results
“Comparative results of SCID NBS are shown in table 2. After reducing our retest cutoff from 34 to 24 copies/μL…” - what was the rationale for picking 24 copies/ul and not lower?
We decided to use this cutoff (1st percentile) in base to the retest rate obtained (similar than publish by others (10)). We didn’t decide a lower cutoff because our cutoff for positive detections was 20 copies/µL. We know our program is conservative, but we didn’t have enough experience with SCID NBS and we wanted to have 5 years data to recalculate both retest and detection cutoff in order to reduce them even more (we think about 15 and 10 copies/µL respectively). We have added this information.
Table 2. – What was the delay between the 1st result and the repeat sample arriving and being tested?
We currently have the results for 2nd samples within 7 following days after request them. We haven’t included this information in the paper.
Page 7 – “There have been three cases of SCID with confirmed diagnosis…” - what were the TREC levels in these 3 cases? What was the median/range TREC levels for the other screen positive groups?
TREC 0, 0 and 4 copies/µL respectively were the values obtained in the three positive cases. TREC 10 [5-18] copies/µL was the median and range for the non-SCID linphopenia and 13 [8-20] copies/µL for the false positive cases. We have added this information.
What were the limits of detection and quantification for this assay based on the authors results?
As we mention on section 2.3.1 Methodology and algorithm validation, “the method was verified (repeatability, reproducibility, limit of detection, limit of quantification, sensitivity, and specificity) and these parameters were successfully compared with those stated in the kit insert (3)”. All this information can be consulted at reference 3 of the bibliography.
Discussion
Page 7, “After three and a half years’ experience and almost 223,000 newbornsscreened, three SCID patients…” - were these from families with known histories of SCID?
Two SCID cases were from consanguineous Moroccan parents, of which, one of them had a history of a sister who died in Morocco with described symptoms compatible with Omenn-type SCID. We have added this information.
Were any SCID babies missed by screening in Catalonia over the 3.5 years?
To the best of our knowledge, no cases of SCID or clinically significant T cell lymphopaenia have been missed in our screened cohort. We have added this information.
Has the incidence of SCID in Catalonia increased since the start of screening?
To the best of our knowledge, the incidence of SCID in Catalonia same be the same since the start of screening.
Would lowering the cut-offs remove non-SCIDs from the positive screen group? If so, could the authors provide details on the numbers in the positive screen group if different cut-offs had been used? For example if the first cut-off was 15 or 20 copies what impact would this have? Maybe a table with various cut-off alongside referral numbers would be helpful.
We are currently analyzing this very interesting aspect. We are working in the paper to explain our experience after 5 years of neonatal screening of SCID in Catalonia and we intend to include this data.
As we already mentioned, this publication was to summarize the communication made during the virtual meeting “Newborn Screening for SCID ‘State of the Art’” celebrated on 26 and 27 January 2021. We know some information is missing but we are also working on a more complete publication on the 5 years of neonatal screening for SCID in Catalonia and we will take into account all the comments made.
We really appreciate the great work done by the reviewers and hope they understand our reasons for not incorporating some of the proposed changes.
